# Preparation and Adsorption Properties of Nanostructured Composites Derived from Al/Fe Nanoparticles with Respect to Arsenic

**DOI:** 10.3390/nano12183177

**Published:** 2022-09-13

**Authors:** Sergey O. Kazantsev, Aleksandr S. Lozhkomoev, Nikolay G. Rodkevich

**Affiliations:** Institute of Strength Physics and Materials Science, Siberian Branch, Russian Academy of Sciences, 2/4, Pr. Akademicheskii, 634055 Tomsk, Russia

**Keywords:** bimetallic nanoparticle, oxidation, aluminum oxide, iron, nanostructures, composites, arsenic, adsorption

## Abstract

Composite nanostructures containing iron in different forms exhibit a high adsorption capacity with respect to arsenic. The aim of our study was to investigate the adsorption activity of an adsorbent composite prepared by the oxidation of bimetallic Al/Fe nanoparticles under different conditions. Depending on the oxidation conditions, nanostructures with different morphologies in the form of nanosheets, nanoplates and nanorods with different compositions and textural characteristics could be obtained. The nanostructures obtained had a positive zeta potential and were characterized by a high specific surface area: 330 m^2^/g for the AlOOH/FeAl_2_ nanosheets; 75 m^2^/g for the AlOOH/Fe_2_O_3_/FeAl_2_ nanoplates; and 43 m^2^/g for the Al(OH)_3_/FeAl_2_ nanorods. The distribution of an FeAl_2_ intermetallide over the surface of the AlOOH nanostructures led to an increase in arsenic adsorption of 25% for the AlOOH/FeAl_2_ nanosheets and of 34% for the AlOOH/Fe_2_O_3_/FeAl_2_ nanoplates and Al(OH)_3_/FeAl_2_ nanorods. The adsorption isotherms matched most preciously to the Freundlich model. This fact indicated the energy heterogeneity of the adsorbent surface and multilayer adsorption. The nanostructures studied can be used to purify water contaminated with arsenic.

## 1. Introduction

Arsenic is a very toxic and widespread water pollutant in several regions of the world. It is present in natural water and soil, most often in the form of arsenite and arsenate species or As(III) and As(V), respectively [1,2]. In surface water, arsenic is usually present as As(V) in the form of an H_2_AsO_4_^−^ and HAsO_4_^2−^ species, depending on the water pH value [2].

The main methods of arsenic removal from water are precipitation/coagulation, membrane filtration, ion exchange and adsorption [1,2,3]. Adsorption is considered to be the most effective and easy to use method for arsenic removal from water [1,2,3,4,5,6,7]. Iron oxide and iron oxide-based sorbent mixtures are currently considered to be effective adsorbents for arsenic species [3,8,9,10,11,12,13,14,15,16,17,18,19]. This is due to the affordability and availability of iron oxide-based sorbents as well as the high affinity of iron for arsenic. Iron oxides and hydroxides chemically obtained from ferric salts are able to adsorb arsenic from aqueous solutions quite effectively [6]. The maximum adsorption capacity of such sorbents is 148.7 mg/g. The preparation of a mesoporous iron oxide sorbent with a high specific surface area of 269 m^2^/g has been reported and was capable of adsorbing up to 90% of the arsenic compounds within 5 min [10]. The maximum adsorption capacity of the iron oxide sorbent was about 80 mg/g. The studies of the sorption characteristics of an α-Fe_2_O_3_-based spherically shaped sorbent with a high specific surface of ~162 m^2^/g with respect to As(III) and As(V) showed the sorption capacity to be at least 95 mg/g and 47 mg/g, respectively [8].

To obtain versatile adsorbents for the removal of heavy metal ions, organic compounds and bacteria, for example, in addition to arsenic compounds and composite materials doped with iron species have been created [1,12,13,14,15,16,17,18].

Composite materials based on iron-doped activated carbon [13] have a high specific surface area equal to 998 m^2^/g; the maximum As(V) sorption capacity is 32.57 mg/g. Biochar–iron composites, depending on the method of production, can have a specific surface area from 28.9 to 482.4 m^2^/g [14,15] with a maximum As(V) sorption capacity up to 868 mg/g [15]. The reported As(V) sorption capacity of chitosan-based adsorbents [16] was about 15 mg/g. Hollow polymethylmethacrylate spheres doped with iron oxide with a specific surface area of 8.6 m^2^/g showed an As(V) sorption capacity of 10.031 mg/g [17].

Alumina is considered to be the most promising material for creating composite sorbents due to its enhanced adsorption properties with respect to organic dyes, organic compounds [19,20,21], bacteria and viruses [22,23,24].

The formation of AlOOH/AlFe_2_ flower-like composite nanoparticles by the water oxidation of bimetallic Al/AlFe nanoparticles produced by an electrical explosion of wires has recently been reported [25]. The resulting composites had a high specific surface area and an adsorption capacity exceeding 200 mg/g with respect to As(V). Depending on the oxidation conditions of the bimetallic nanoparticles, nanostructured oxide particles with various morphologies, compositions and textural characteristics could be obtained [26,27]. However, the oxidation behavior of binary Al/AlFe nanoparticles influencing the structural characteristics of the resulting nanocomposites and their adsorption properties have not previously been considered.

In this connection, the aim of this work was to study the features of composite particle formation during the water oxidation of binary Al/AlFe nanoparticles under different conditions and to study their adsorption properties with respect to As(V).

## 2. Materials and Methods

Bimetallic Al/Fe nanoparticles were obtained by an electrical explosion of twisted aluminum and iron wires in an argon atmosphere according to the method described previously [28]. The metal ratio in the obtained nanoparticles could be varied by changing the diameter of the wires used. In the present work, the diameter of the Fe wire was 0.1 mm and the diameter of the Al wire was 0.35 mm, providing an iron/aluminum ratio of 10:90 by weight.

To obtain the composite adsorbents, bimetallic Al/Fe nanoparticles were oxidized in three ways: with water at 60 °C for 1 h; in humid air at 80% relative humidity at 60 °C for 72 h; and in hydrothermal conditions at 200 °C for 6 h. These conditions were chosen according to previous studies [28], which showed the possibility of the formation of flower-like, nanorod-like and nanosheet aluminum oxide nanoparticles.

The resulting samples were characterized by X-ray diffraction (XRD) in a Shimadzu XRD 6000 diffractometer operating with Cu Kα. A qualitative phase analysis was carried out with a Powder Diffraction File (PDF) database, PDF-2 Release 2014. The structure of the samples was studied by transmission electron microscopy (TEM) using a JEM-2100 electron microscope (JEOL, Tokyo, Japan). The distribution of elements in the particles was evaluated using the X-Max energy dispersion analysis system, EDS, integrated into the microscope (Oxford Instruments, Oxfordshire, Abingdon, UK). Zeta potential measurements were performed using a Zetasizer Nano ZSP instrument (Malvern Instruments Ltd., GB, Malvern, UK) with Zetasizer software. The surface area of the samples and pore structure were measured by nitrogen adsorption/desorption using a Sorbtometer M (Katakon, Novosibirsk, Russia) automatic analyzer.

The adsorption properties of the synthesized nanostructures with respect to the As(V) ions were studied by the batch technique. A stock solution of sodium arsenate (500 mg/mL in terms of As(V)) was prepared; other solutions were prepared by a subsequent dilution. To construct the adsorption isotherm curve, a 50 mg adsorbent sample was placed in 50 mL of a sodium arsenate solution with varying concentrations ranging from 20 to 500 mg/L followed by constant stirring for 60 min. The adsorption capacity *q_e_* (mg/g) was calculated using Equation (1):(1)$$q_{e}=\frac{\left(C_{0}-C_{e}\right)\cdot V}{m}$$
where *C*_0_ and *C_e_* are the initial and equilibrium concentrations of the solute, respectively, *V* is the solvent volume and *m* is the adsorbent mass.

The arsenic concentration was measured by the inversion voltammetry method, pre-reducing As(V) to As(III) using a TA-Lab voltammetric analyzer (TomAnalit, Tomsk, Russia). A carbon–gold electrode was used as a measuring electrode and silver chloride was used as a reference electrode with 0.01 mol/L Trilon B as the background electrolyte. The method detection limit was 0.5 µg/L.

## 3. Results and Discussion

According to the TEM data, the studied bimetallic Al/Fe nanoparticles had a spherical shape and contained both Al and Fe, as could be seen from the EDS data (Figure 1a). The composition of the bimetallic Al/Fe nanoparticles corresponded with the theoretical one; the iron content determined by the TEM-EDS method was 11.8 wt. % (Figure 1b). The studied nanoparticle particle size distribution obeyed the log-normal function; the average particle size was 93 nm (Figure 1c).

Figure 2 shows the XRD pattern of the bimetallic Al/Fe nanoparticles. As can be seen, high intensity peaks characteristic of Al with a standard lattice parameter (4.049 Å) and reflexes characteristic of an FeAl_2_ intermetallide were observed. The presence of a large amount of free aluminum in the nanoparticles contributed to the formation of aluminum oxyhydroxides and hydroxides when interacting with water, as has previously been shown [27].

Depending on the reaction conditions of the water interaction with the bimetallic Al/Fe nanoparticles, nanostructures with different morphologies were formed. Interacting in water at 60 °C, agglomerates were formed; these were up to 2 µm in size and consisted of nanosheets 2–5 nm thick and up to 200 nm wide. Nanosheets up to 150 nm in size and up to 10 nm in thickness were formed by a hydrothermal treatment. Rod-like nanoparticles up to 80 nm long and up to 100 nm in diameter were formed by oxidation in humid air at 80% relative humidity and a temperature of 60 °C. The morphology of nanoparticles formed during the oxidation of bimetallic Al/Fe nanoparticles slightly differs from that of nanoparticles formed during the oxidation of Al nanoparticles under similar conditions [28]; the FeAl_2_ intermetallide, which was present in the bimetallic nanoparticles, had no noticeable effect on the oxidation process. According to the TEM-EDS elemental analysis, iron was evenly distributed throughout the particles and the iron/aluminum ratio corresponded with that of the initial nanoparticles (Figure 3).

Figure 4 shows the XRD pattern of the water oxidation products. On the XRD diffractogram of the nanosheet structures, broadened peaks characteristic of fine crystalline boehmite AlOOH as well as peaks of the FeAl_2_ intermetallide were observed (Figure 4a). The nanosheets obtained under hydrothermal conditions had intense peaks of boehmite, the FeAl_2_ intermetallide and ferric oxide Fe_2_O_3_ (Figure 4b). The formation of boehmite with a high degree of crystallinity and Fe_2_O_3_ was due to more harsh reaction conditions (elevated pressure and a temperature of 200 °C) in which the intermetallide partially decomposed and boehmite crystallization occurred. The composition of the rod-like nanoparticles formed in humid air was represented by the bayerite Al(OH)_3_ and FeAl_2_ phases (Figure 4c).

Figure 5 shows the N_2_ adsorption–desorption isotherms for the synthesized nanostructures. According to the IUPAC classification, the shapes of the isotherms of the nanosheets and nanoplates corresponded with type IV, which characterized the presence of slit-shaped pores in the samples. The shape of the isotherms of the rod-like nanoparticles corresponded with type III, characteristic of non-porous samples. The N_2_ adsorption–desorption isotherms of the AlOOH/FeAl_2_ nanosheets showed pronounced hysteresis, indicating a mesoporous structure of the sample. The specific surface area was 330 m^2^/g (Figure 5a, curve 1). It should be noted that during the water oxidation of the Al/Fe nanoparticles containing 50 wt % Fe, the specific surface area of the resulting nanostructures reached 247 m^2^/g [25], which may have been due to the presence of larger Fe-based particles in the reaction products. The reduction of iron in the initial nanoparticles allowed the nanosheet structures with a high specific surface area to be determined not only by the surface of the boehmite nanosheet, which had a typical specific surface area of about 250 m^2^/g [27], but also by the fine iron-based particles dispersed on the nanosheet surface. The AlOOH/Fe_2_O_3_/FeAl_2_ nanoplates had a mesoporous structure with a pronounced maximum on the pore size distribution curve in the 4 nm region (Figure 5b, curve 1). For the boehmite nanoplates, the hysteresis was weakly pronounced and the value of the specific surface area was 75 m^2^/g (Figure 5a, curve 2). The pore size distribution indicated the presence of mesopores with a size of 5–20 nm (Figure 5b, curve 2). In spite of the nature of the N_2_ adsorption–desorption isotherms of the Al(OH)_3_/FeAl_2_ nanorods, the BET surface area was 43 m^2^/g (Figure 6a, curve 3), which was determined by the outer surface of the particles.

One of the important characteristics of the adsorbents was the surface charge, which could be evaluated as the zeta potential of the adsorbent surface. The obtained nanostructures had positive zeta potential values in water in a pH range from 3 to 9, which should have contributed to the electrokinetic interaction of the sorption sites with the arsenic ions (Figure 6). It should be noted that the zeta potential of the synthesized nanostructures was predominantly determined by the properties of the boehmite nanosheets. Similar dependencies were established for the nanostructures synthesized by the water oxidation of the Al-based nanoparticles [27]. The increase in the iron content of the nanostructures led to a decrease in the zeta potential in the region of physiological pH values of 25–30 mV, which was also previously observed for the Al/Fe nanoparticles containing 50 wt% Fe (zeta potential in the pH 7–8 range was 15–20 mV) [25]. Thus, increasing the amount of Fe in the precursor (Al/Fe) negatively affected its electrokinetic properties.

The adsorption isotherms of the As(V) ions on the surface of the synthesized samples are shown in Figure 7. For a comparison, Figure 7b shows the adsorption isotherms of the As(V) ions on the surface of the nanostructures obtained by the water oxidation of the Al nanoparticles without Fe moieties under similar conditions. The maximum adsorption capacity observed on the AlOOH/FeAl_2_ nanosheet samples was about 102 mg/g; for the Al(OH)_3_/FeAl_2_ nanorods, it was about 87 mg/g and for the AlOOH/Fe_2_O_3_/FeAl_2_ nanoplates, it was about 77 mg/g.

As can be seen from the adsorption isotherm curves, the nanostructures not containing iron (Figure 7b) had a sufficiently high adsorption capacity with respect to the As(V) ions. The presence of iron in the studied samples led to an increase in the adsorption capacity (Figure 7a).

For a mathematical description of the process of As(V) adsorption, the most common models used are the Langmuir and Freundlich models [29,30,31,32,33]. The model availability was evaluated using determination coefficients (*R*^2^) [33].

The Langmuir model assumes that a sorbate monolayer is adsorbed on a homogeneous adsorbent surface at a constant temperature and that the distribution of the sorbate between the two phases is evaluated by the equilibrium constant. Hence, the adsorption and desorption rates are equal at an equilibrium state. The equation describing the Langmuir model has the form:(2)$$q_{e}=q_{max}\;\frac{K_{a}C_{e}}{1+K_{a}C_{e}}$$
where *q_e_* (mg/g) is the amount of As(V) sorption, *C_e_* (mg/L) is the concentration at the equilibrium state, *K_a_* is the Langmuir constant (L/mg) and *q_max_* is the monolayer Langmuir capacity (mg/g).

The linearized form of the Langmuir equation is:(3)$$\frac{1}{q_{e}}=\left(\frac{1}{K_{a\;}q_{max\;}}\right)\frac{1}{C_{e}}+\left(\frac{1}{q_{max}}\right)$$

The Freundlich model assumes that the adsorption process occurs on an inhomogeneous surface. The Freundlich model equation is:(4)qe=KfCe1n
where *K_f_* (L/mg) is an indicator of the adsorption capacity and 1/*n* is the adsorption intensity, indicating both the relative distribution of the energy and the heterogeneity of the adsorbent sites.

The linearized form of the Freundlich adsorption model is:(5)$$\log q_{e}=\log K_{f}+\frac{1}{n}logC_{e}\;$$

Figure 8 shows the Langmuir and Freundlich adsorption isotherms presented in a linearized form. Based on the *R*^2^ coefficient values, the As(V) adsorption for all types of nanostructures synthesized from the Al/Fe nanoparticles was described most adequately by the Freundlich equation. This indicated that the surface of the synthesized sorbents was energetically heterogeneous and that multilayer adsorption was likely. The high value of the *R*^2^ coefficient for the Freundlich model indicated that the surface of the studied sorbent contained active centers with a different strength.

In addition, the value of 1/*n* (the Freundlich isotherm constant) could be used to calculate the adsorption capacity and adsorption rate. In the case of 1/*n* = 0, the sorption process was irreversible; in the case of 0 < 1/*n* < 1, it was favorable; and in the case of 1/*n* > 1, it was unfavorable [34]. According to Table 1, the values of 1/*n* were in a range from 0 to 1 in all cases with the 1/*n* value being the maximum for the nanoplates.

Thus, the presence of iron moieties in the nanostructures led to an increase in the adsorption capacity with respect to As(V) of 25–34%. The nanoplates were characterized by the lowest sorption capacity with respect to As(V) and the highest value of 1/*n*, which indicated a weak adsorbate/adsorbent interaction compared with the other nanostructures. It should also be noted that despite significant differences in the specific surface area, the synthesized nanostructures adsorbed the As(V) ions to the same extent. The sorption capacity of the nanosheets was 0.31 mg/m^2^, the sorption capacity of the nanoplates was 0.97 mg/m^2^ and the sorption capacity of the nanorods was 2.02 mg/m^2^ per surface unit. In this connection, it could be assumed that arsenic adsorption predominantly occurred on the active surface centers of the nanostructures, the number of which for the synthesized ones had a close value whilst their concentration on the surface differed or a significant part of the porous space of the nanosheets was not available for As(V) adsorption.

## 4. Conclusions

It was found that, depending on the conditions of oxidation of bimetallic Al/Fe nanoparticles, it was possible to obtain nanostructures with morphologies of nanosheets, nanoplates and nanorods with different textural characteristics and compositions. As a result of nanoparticle oxidation, iron moieties, mainly in the form of an FeAl_2_ intermetallide, were uniformly distributed on the surface of the nanostructures, which led to an increase in As(V) adsorption of ~25% for the nanostructures oxidized in water and of 34% for the nanostructures oxidized in humid air and under hydrothermal conditions.

The efficiency of As(V) adsorption was shown to weakly depend on the specific surface area of the synthesized nanostructures and was probably determined by the concentration of the active adsorption sites responsible for arsenic adsorption. In all cases, the experimental adsorption curves were best described by the Freundlich model, which indicated the heterogeneity of the nanostructure surface and the probability of multilayer adsorption. The maximum adsorption capacity was characteristic of particles in the form of nanosheets obtained by water oxidation at 60 °C on Al/Fe particles and was 102 mg/g.

## Figures and Tables

**Figure 1 nanomaterials-12-03177-f001:**
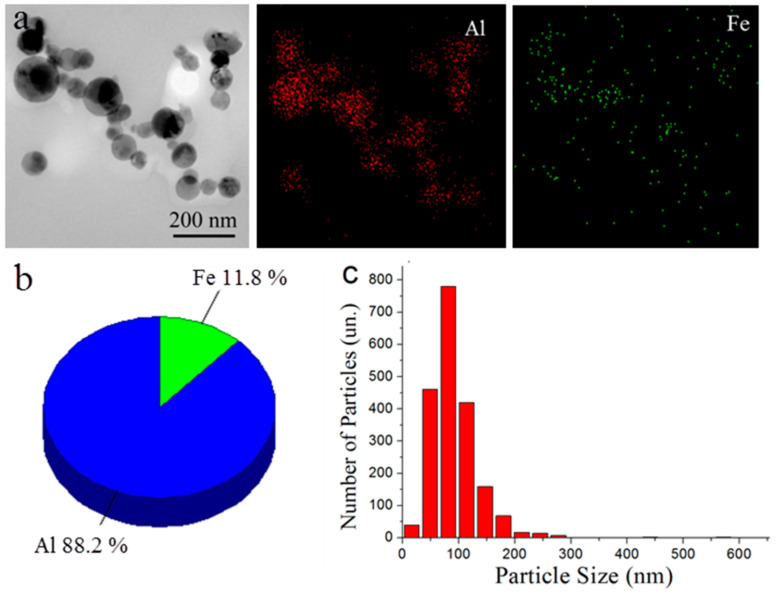
TEM images and elemental distribution maps (**a**), elemental mass distribution (**b**) and size distribution curve (**c**) of Al/Fe nanoparticles.

**Figure 2 nanomaterials-12-03177-f002:**
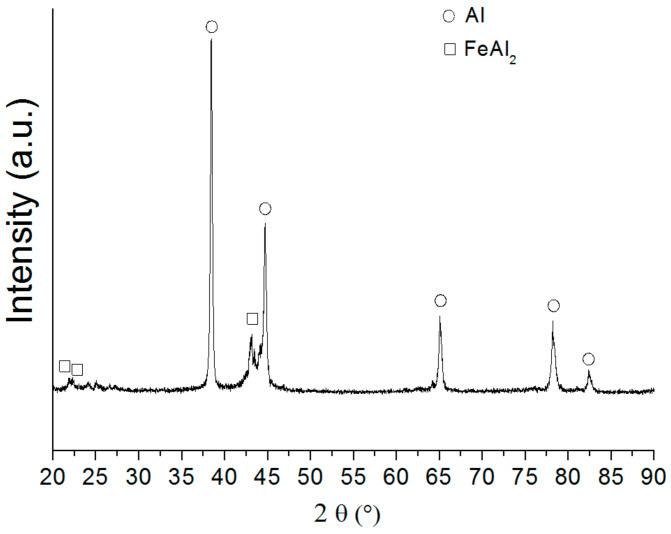
XRD pattern of the bimetallic Al/Fe nanoparticles.

**Figure 3 nanomaterials-12-03177-f003:**
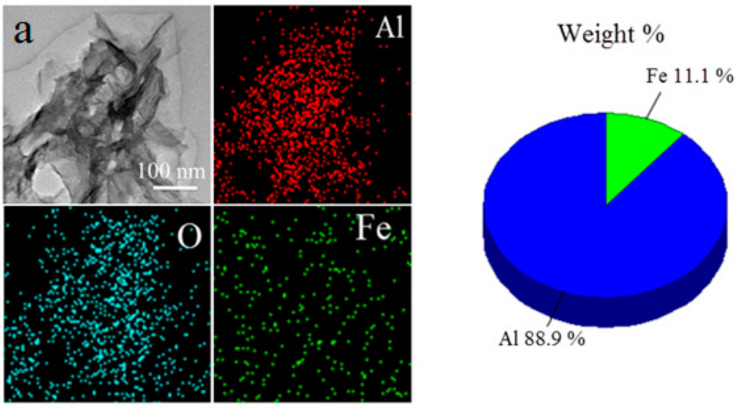
Interaction products of the bimetallic Al/Fe nanoparticles with water and Al/Fe ratio under different conditions: (**a**) in liquid water at 60 °C; (**b**) under hydrothermal conditions at 200 °C; and (**c**) in humid air at 60 °C.

**Figure 4 nanomaterials-12-03177-f004:**
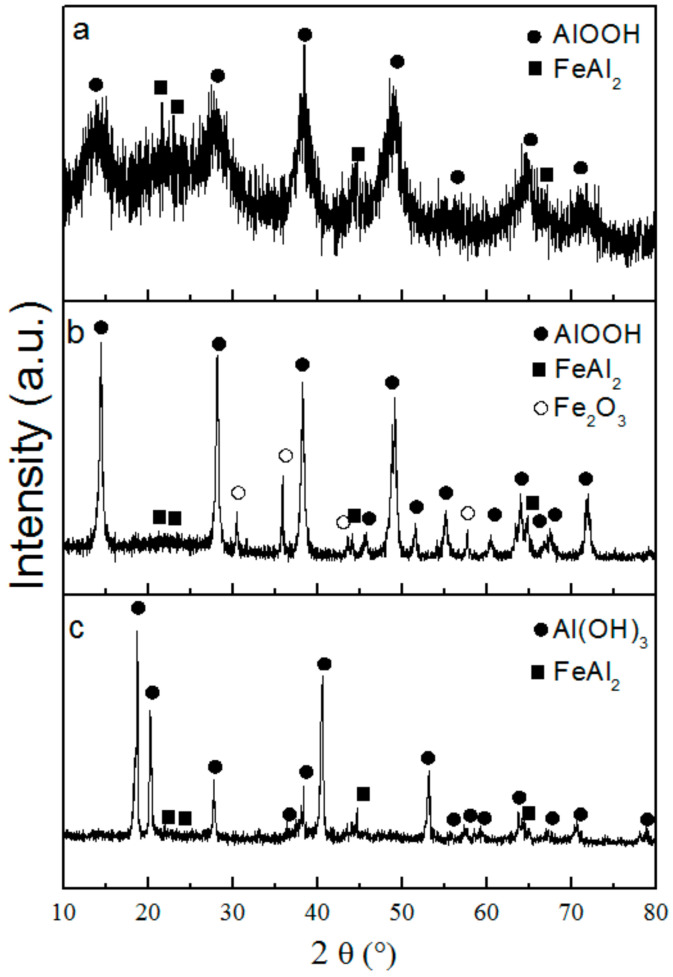
XRD patterns of nanosheets (**a**), nanoplates (**b**) and nanorods (**c**).

**Figure 5 nanomaterials-12-03177-f005:**
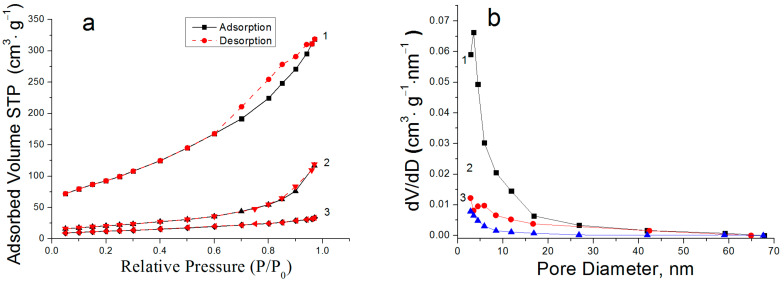
N_2_ adsorption–desorption isotherms (**a**) and pore size distribution curves (**b**) of nanosheets (1), nanoplates (2) and nanorods (3).

**Figure 6 nanomaterials-12-03177-f006:**
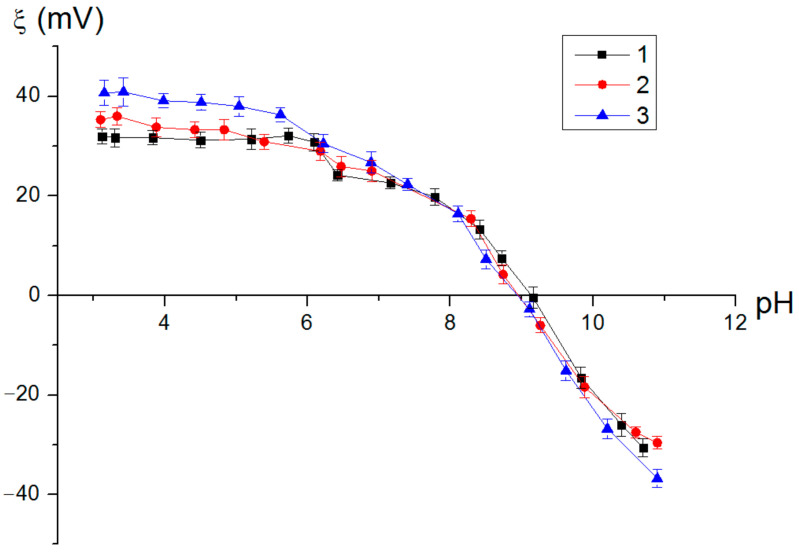
Dependence of the zeta potential value of nanosheets (1), nanoplates (2) and nanorods (3) on the pH value of the medium.

**Figure 7 nanomaterials-12-03177-f007:**
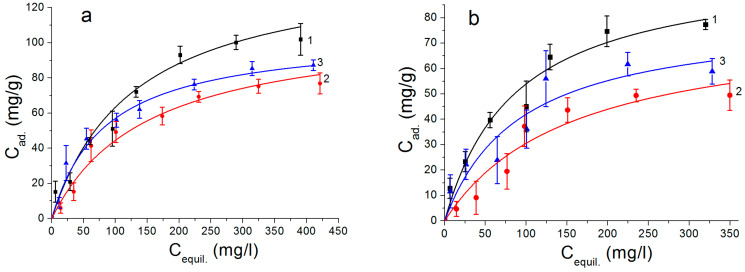
Adsorption isotherms of As(V) ions on the nanostructures with (**a**) and without iron (**b**). 1: Nanosheets; 2: nanoplates; 3: nanorods.

**Figure 8 nanomaterials-12-03177-f008:**
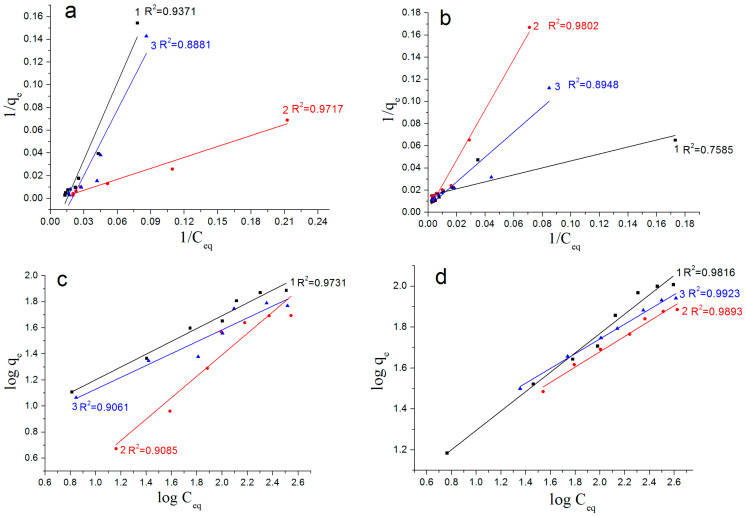
Linearized Langmuir (**a**,**b**) and Freundlich (**c**,**d**) adsorption isotherms for Al and Al/Fe nanostructures obtained by the water oxidation of Al (**a**,**c**) and Al/Fe (**b**,**d**) nanopowders. 1: Nanosheets; 2: nanoplates; 3: nanorods.

**Table 1 nanomaterials-12-03177-t001:** The Langmuir and Freundlich model parameters.

Samples	BET Surface Areas (m^2^/g)	Langmuir Model			Freundlich Model		
		*q_max_* (mg/g)	*K_a_*	*R* ^2^	*K_f_*	1/*n*	*R* ^2^
AlOOH nanosheets	252	80.5	0.0201	0.9371	5.17	0.48	0.9731
AlOOH nanoplates	72	117.3	0.0027	0.9717	0.58	0.79	0.9085
Al(OH)_3_ nanorods	60	54.7	0.0259	0.8881	4.68	0.45	0.9061
AlOOH/FeAl_2_ nanosheets	330	135.2	0.0080	0.7585	4.57	0.53	0.9816
AlOOH/Fe_2_O_3_/FeAl_2_ nanoplates	75	141.1	0.0037	0.9802	1.01	0.75	0.9893
Al(OH)_3_/FeAl_2_ nanorods	43	116.0	0.0089	0.8948	2.53	0.62	0.9923

## Data Availability

Not applicable.

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
