# Peer review of "Preparation and Adsorption Properties of Nanostructured Composites Derived from Al/Fe Nanoparticles with Respect to Arsenic"

_nanomaterials, 2022, doi:10.3390/nano12183177_

Round 1

Reviewer 1 Report

This paper investigated the influence of oxidized binary Al/Fe nanopartiticles on the As adsorption behavior. The analytical methods, results, and conclusions were well-organized and presented in the paper. Overall, this paper can be accepted before: (1) some discussion should be added regarding to the different adsorption characteristics of the different nanoparticles, (2) for each figures the error bar should be added, and (3) written English mistake should be corrected.

Author Response

Comments and Suggestions for Authors

This paper investigated the influence of oxidized binary Al/Fe nanopartiticles on the As adsorption behavior. The analytical methods, results, and conclusions were well-organized and presented in the paper. Overall, this paper can be accepted before: (1) some discussion should be added regarding to the different adsorption characteristics of the different nanoparticles, (2) for each figures the error bar should be added, and (3) written English mistake should be corrected.

Response:

(1) We have added a discussion of the adsorption characteristics of various nanoparticles to Section I.

(2) We have added error bars to the figure where possible.

(3) Errors have been corrected

Reviewer 2 Report

In this manuscript, the authors prepared Al/Fe composite nanomaterials with different structures and experimentally verified the adsorption of arsenic. This work sounds very interesting and interesting, and the analysis is relatively clear, but there are still some concerns regarding this paper. Therefore, I suggest to publish it in the journal with significant revisions:

1. I suggest that the authors adjust the content of the abstract by reducing the description of the preparation method and adding a conclusion section.

2. The rationale for choosing nanoparticles containing 10 wt. % Fe and 90 wt. % Al is not clearly stated in the text, and whether there is a significant effect on the adsorption performance when the ratios are different. Please try more analysis.

3. Please provide literature support for some of the explanations in the text, for example, the analysis that the maximum adsorption value does not depend on the specific surface area of the nanostructure.

4. The article is more of a description of the phenomenon and lacks theoretical analysis, so please make additions.

5. There have some grammatical mistakes and typos. Therefore, the manuscript should be polished carefully before publication.

Author Response

We thank the reviewer for the effort and time put into the review of the manuscript. Each comment has been carefully considered point by point and responded.

Comments and Suggestions for Authors

In this manuscript, the authors prepared Al/Fe composite nanomaterials with different structures and experimentally verified the adsorption of arsenic. This work sounds very interesting and interesting, and the analysis is relatively clear, but there are still some concerns regarding this paper. Therefore, I suggest to publish it in the journal with significant revisions:

  1. I suggest that the authors adjust the content of the abstract by reducing the description of the preparation method and adding a conclusion section.

Response: we have corrected the content of the Abstract in accordance with the comment.

  1. The rationale for choosing nanoparticles containing 10 wt. % Fe and 90 wt. % Al is not clearly stated in the text, and whether there is a significant effect on the adsorption performance when the ratios are different. Please try more analysis.

Response:  In this case, we considered a system of 10 wt% Fe and 90 wt% Al to improve the adsorption characteristics of AlOOH nanoparticles formed by oxidation of Al nanoparticles with respect to arsenic compounds. In our previous works, nanostructures derived from Al/Fe nanoparticles with 50 wt% Fe and 50 wt% Al were investigated and showed higher adsorption values with respect to As(V), but the anionic dye removal efficiency and bacterial trapping. We have added more analysis to the manuscript text on this subject.

  1. Please provide literature support for some of the explanations in the text, for example, the analysis that the maximum adsorption value does not depend on the specific surface area of the nanostructure.

Response: In this case, the synthesized nanostructures were meant. Despite a significant difference in the specific surface value, the sorption capacity of the samples remained at the same level. We have corrected the manuscript in accordance with the remark and introduced literature confirmation of some statements made.

  1. The article is more of a description of the phenomenon and lacks theoretical analysis, so please make additions.

Response:  We have made appropriate additions to the manuscript.

  1. There have some grammatical mistakes and typos. Therefore, the manuscript should be polished carefully before publication.

Response:  Spelling and grammatical errors have been corrected.

Reviewer 3 Report

Comments from Reviewer

Title: Preparation and adsorption properties of nanostructured composites derived from Al/Fe nanoparticles with respect to arsenic

The current form's presentation of methods and scientific results is satisfactory for publication in the Nanomaterials journal. The minor and significant drawbacks to be addressed can be specified as follows:
1.    Page 1, Abstract. "is m2/g;"??? Probably the value is omitted.
2.    2. Materials and Methods. ---> 2. Materials and methods.
3.    Fig. 1. (i) Number of particles ---> Number of Particles (ii) Particle size ---> Particle Size. See Fig. 3 and tittles of axes.
4.    Page 4. "show pronounced hysteresis, which indicates high BET surface area being". That's not true. The C-point on the adsorption isotherm is responsible for the surface area. It can be equated with the inflection point. The hysteresis loop is related to the nature of the mesopores.
5.    Fig. 5, curve 3. The material is non-porous.
6.    Fig. 5, y-axis. Volume? Volume --> adsorption.
7.    Fig. 5. Values of the adsorbed amount should be expressed in cm3 STP g-1 !!!!!!!!!!!!! See y-axis!!!!!!!!!!  From the analysis of the adsorption values, it is seen that the unit is cm3 STP/g (or cm3/g STP). It is a typical error. The values of nitrogen adsorption for different adsorbents are typically below 1-2 cm3/g!!!!
=============================================================
STP - Abbreviation for standard temperature (273.15 K or 0 °C) and pressure (105
        Pa); usually employed in reporting gas volumes. Note that flow meters calibrated
        in standard gas volumes per unit time often refer to volumes at 25 °C, not 0 °C.
8.    Fig. 5, PSD. The BJH method? What the shape of pores has been assumed.
9.    Fig. 7. Why are the two panels different sizes? Differences in the fonts. As such, this drawing is unacceptable.
10.    References. There were numerous typographical errors. - lack of attention to detail, consistent with the standard of a scientific presentation. Literature should also be standardized: the size of letters in the titles of journals, initials of names, and the size of letters in the titles of articles.

Sincerely,

    The reviewer.

Author Response

We thank the reviewer for the effort and time put into the review of the manuscript. Each comment has been carefully considered point by point and responded. The changes in the text are highlighted.

Comments and Suggestions for Authors

The current form's presentation of methods and scientific results is satisfactory for publication in the Nanomaterials journal. The minor and significant drawbacks to be addressed can be specified as follows:

  1. Page 1, Abstract. "is m2/g;"??? Probably the value is omitted.

Response:  We have corrected the manuscript accordingly.

  1. 2. Materials and Methods. ---> 2. Materials and methods.

Response:  We have corrected the manuscript accordingly.

  1. Fig. 1. (i) Number of particles ---> Number of Particles (ii) Particle size ---> Particle Size. See Fig. 3 and tittles of axes.

Response:  We have corrected the manuscript accordingly.

  1. Page 4. "show pronounced hysteresis, which indicates high BET surface area being". That's not true. The C-point on the adsorption isotherm is responsible for the surface area. It can be equated with the inflection point. The hysteresis loop is related to the nature of the mesopores.

Response:  We have corrected the manuscript accordingly.

  1. Fig. 5, curve 3. The material is non-porous.

Response:  We have corrected the manuscript accordingly.

  1. Fig. 5, y-axis. Volume? Volume --> adsorption.

Response:  Axes names have been corrected.

  1. Fig. 5. Values of the adsorbed amount should be expressed in cm3 STP g-1 !!!!!!!!!!!!! See y-axis!!!!!!!!!!  From the analysis of the adsorption values, it is seen that the unit is cm3 STP/g (or cm3/g STP). It is a typical error. The values of nitrogen adsorption for different adsorbents are typically below 1-2 cm3/g!!!!
    =============================================================
    STP - Abbreviation for standard temperature (273.15 K or 0 °C) and pressure (105
         Pa); usually employed in reporting gas volumes. Note that flow meters calibrated
            in standard gas volumes per unit time often refer to volumes at 25 °C, not 0 °C.

Response:  Axis format has been corrected.

  1. Fig. 5, PSD. The BJH method? What the shape of pores has been assumed.

Response:  The pore size distribution is plotted using the BJH method, which uses a cylindrical pore model. The character of the adsorption-desorption isotherms of N2 curve 1 and curve 2 (Fig. 5 a) corresponds to slit-shape pores..

  1. Fig. 7. Why are the two panels different sizes? Differences in the fonts. As such, this drawing is unacceptable.

Response:  Axis format has been corrected.

  1. References. There were numerous typographical errors. - lack of attention to detail, consistent with the standard of a scientific presentation. Literature should also be standardized: the size of letters in the titles of journals, initials of names, and the size of letters in the titles of articles.

Response:  We have corrected the presentation of the text in accordance with the remark. The reference list has also been corrected.

Round 2

Reviewer 1 Report

The paper can be accepted. 

Reviewer 2 Report

It can be accepted as it is.

Reviewer 3 Report

In my opinion the corrected work can be accepted for publication.